# Effectiveness of Acupuncture on Health-Related Quality of Life in Patients Receiving Maintenance Hemodialysis

**DOI:** 10.3390/healthcare11091355

**Published:** 2023-05-08

**Authors:** Marta Correia de Carvalho, José Nunes de Azevedo, Pedro Azevedo, Carlos Pires, Jorge Pereira Machado, Manuel Laranjeira

**Affiliations:** 1ICBAS, School of Medicine and Biomedical Sciences, University of Porto, 4050-313 Porto, Portugal; 2TECSAM-Tecnologia e Serviços Médicos SA, 5370-530 Mirandela, Portugal; 3Center for Research in Neuropsychology and Cognitive and Behavioral Intervention (CINEICC), Faculty of Psychology and Educational Sciences, University of Coimbra, 3000-115 Coimbra, Portugal; 4CBSin, Center of BioSciences in Integrative Health, 4000-105 Porto, Portugal; 5INC, Instituto de Neurociências, 4100-141 Porto, Portugal

**Keywords:** chronic kidney disease, hemodialysis, acupuncture, health-related quality of life, integrative medicine, randomized controlled trial

## Abstract

Patients with kidney failure (KF) receiving maintenance hemodialysis (HD) experience numerous symptoms that impair their health-related quality of life (HRQOL) and contribute to high mortality rates. Acupuncture is often used for symptom enhancement and HRQOL. This blinded, randomized, controlled patient-assessor trial evaluated the effectiveness of acupuncture compared with sham acupuncture on patients’ HRQOL receiving maintenance HD as a secondary analysis. Seventy-two participants were randomly assigned to verum acupuncture (VA), sham acupuncture (SA), or waiting-list (WL) groups. The outcome was an improvement in HRQOL, assessed using the Kidney Disease Quality of Life—Short Form, version 1.3 (KDQOL-SF™ v1.3) at baseline, after treatment, and at 12-week follow-up. Non-parametric tests were used for statistical analysis. Of the 72 randomized patients, 67 were included in the complete analysis set. As for the changes between baseline and after treatment, the VA group showed significantly increased scores on most of the KDQOL-SF™ v1.3 scales compared to SA or WL groups (*p* < 0.05). No statistically significant differences between groups were observed in the changes from baseline to follow-up (*p* > 0.05). Compared to the sham treatment, acupuncture improved the HRQOL in patients receiving maintenance HD after treatment but not at follow-up.

## 1. Introduction

The increasing number of chronic kidney disease (CKD) patients with category five glomerular filtration rates (CKG G5) is considered a public health concern [1]. The latest data from CaReMe CKD study indicates that the prevalence of CKD is around 10% in the adult population, usually underestimated in most studies, with substantial mortality rates and high health costs [2]. In Portugal, the prevalence of CKD is high compared to other countries, and its diagnostic and therapeutic approaches are more expensive than other very prevalent chronic diseases, such as heart failure, acute myocardial infarction, or peripheral arterial disease [2,3].

CKD is a debilitating disease, and its progression often requires kidney replacement therapy (KRT), such as hemodialysis (HD) treatment.

Patients in maintenance HD have a complex treatment and experience numerous physical and emotional limitations that severely impact their daily living activities and clinical outcomes [4,5,6]. While innovation and improvements in technologies associated with HD treatment have contributed to an increased life expectancy, patients often experience physical and mental disabilities and a poor HRQOL [5,7,8]. 

Although difficult to define given the multiple variables inherent in its meaning, HRQOL is considered an individual’s subjective assessment of the impact of their health status on various aspects of their life over time. It includes physical, mental, emotional, and social conceptual dimensions [8,9]. 

Previous studies have consistently reported a significant reduction in HRQOL measures in the patients undergoing HD compared to the clinical and non-clinical populations [9,10,11,12,13]. An association between reduced HRQOL scores and adverse events, such as hospitalization and mortality, has been demonstrated [10,14,15]. Regarding HRQOL component summary scores, recent research reported that a lower physical component summary (PCS) three months after the initiation of dialysis significantly correlates with overall mortality rates in patients undergoing HD [16]. Comorbid conditions of depression and diabetes also predicted a decreased PCS [17].

The importance of the HRQOL concept has increased as a measure to assess treatment outcomes and monitor the quality of care delivered to maintenance dialysis patients. KDOQI recommendations emphasize the relevance of measuring HRQOL in assessing patients’ well-being and the adequacy of new therapeutic approaches [18].

Complementary and integrative medicine interventions have been reported to positively impact HRQOL and well-being in different clinical populations and with different medical conditions [19,20]. Acupuncture, as a traditional Chinese medicine (TCM) practice, has become progressively more accepted and is often used for pain management, supportive care for oncological diseases [21], symptom enhancement for chronic illnesses [22,23,24,25], and HRQOL [26,27]. 

The literature reports that acupuncture may be a feasible and safe add-on treatment option for managing symptoms and improving HRQOL in patients undergoing regular HD [28,29,30]. The study conducted by Jung et al. (2022) shows promising results by demonstrating the ability for acupuncture to potentially preserve residual renal function in HD patients, as it increased their residual urine volume and glomerular filtration rate after eight sessions of interdialytic acupuncture [31]. Additionally, positive effects of acupuncture on functional capacity and peripheral muscle strength in patients undergoing HD were reported [32]. Furthermore, a recent systematic review and meta-analysis assessed the effectiveness and safety of acupuncture in treating uremic pruritus among patients undergoing HD. The analysis concluded that combining acupuncture with HD was more effective than hemodialysis alone in alleviating symptoms of uremic pruritus [33].

To overcome the lack of randomized clinical trials (RCT) evaluating the effectiveness of acupuncture on the HRQOL of patients receiving maintenance HD, this study presents new data not described in the primary analysis of a previously conducted RCT [32,34]. A new analysis was performed to verify the effectiveness of acupuncture on HRQOL improvement, the specific effect of acupuncture compared to placebo, and its short-term effects.

## 2. Materials and Methods

### 2.1. Study Design and Participants

The present study is a new and secondary analysis of a randomized controlled trial [32,34] conducted at a Portuguese hemodialysis center. This study began in August 2021 and ended in February 2022, and was designed to assess the effectiveness of acupuncture on health-related quality of life in advanced CKD patients with kidney replacement therapy (KRT). This study received ethical approval from the University Hospital Center of the Porto/ICBAS–School of Medicine and the Biomedical Sciences ethics commission and was registered on the ClinicalTrials.gov platform.

Male and female participants were enrolled who were over eighteen years old, had kidney failure, were on KRT for more than three months with a regular three-weekly HD program lasting four hours, and had stable clinical status.

Exclusion criteria were applied to participants with contraindicated medical conditions, unstable angina pectoris, malignant hypertension, poorly controlled diabetes mellitus, cerebrovascular events and syncope, decompensated heart disease, a severe mental disorder, or a cognitive disability. Additionally, those who declined to participate, who were incapable of performing physical activity, who recently received acupuncture treatment (less than two weeks prior), who had encountered established allergic responses or other adverse effects resulting from prior acupuncture treatment, or were incapable of complying with the necessary actions involved in the procedure were also considered ineligible for the clinical trial.

Before enrollment, every eligible participant who agreed to participate provided written informed consent following the revised version of the Declaration of Helsinki and the Oviedo Convention.

### 2.2. Random Assignment and Masking

Eligible participants were assigned in an equal allocation ratio to either the verum acupuncture group, sham acupuncture group, or waiting-list group using simple randomization procedures. An independent researcher created the randomization sequences using Microsoft^®^ Excel^®^ for Microsoft 365 MSO. The assignments were then placed in indistinct envelopes to ensure allocation series blinding. Regarding the random allocation of participants in the subgroups created according to the frequency of acupuncture treatment, it was determined that the first twelve participants from both the VA and the SA groups would be allocated to subgroup A (three treatments three times a week). The remaining twelve would be allocated to subgroup B (one treatment once a week) [34]. 

In order to ensure blinding, the group allocation was kept concealed from participants, outcome assessors, and the statistician. Additionally, only the TCM practitioner responsible for administering the acupuncture treatments was aware of each group’s specific intervention.

### 2.3. Intervention

The acupuncture intervention was based on a standardized and repeatable protocol developed by the research team. Details of acupuncture treatment were described according to the revised Standards for Reporting Interventions in Clinical Trials of Acupuncture (STRICTA)* 2010 Checklist [35], as presented in a prior study [34]. 

Regarding the acupuncture rationale, acupoints were chosen through a consensus method, which involved reviewing relevant literature, considering the general principles of acupuncture and TCM meridian theory, and drawing upon clinical expertise from specialists in TCM, acupuncture, and nephrology [36,37,38].

Subgroups were defined to assess the impact of acupuncture frequency on HRQOL in both the verum acupuncture (VA) and sham acupuncture (SA) groups. Subgroup A underwent acupuncture treatment three times a week for three weeks (3 × 3), while subgroup B received treatment only once weekly for nine weeks (1 × 9). In total, both subgroups received a total of nine acupuncture sessions. These treatment frequencies (3 × 3 or 1 × 9) were chosen based on the participants’ hemodialysis (HD) routines. Patients at the TECSAM hemodialysis center undergo HD three times a week, making 3 × 3 and 1 × 9 treatment frequencies appropriate for this study. Considering the physical and emotional burden of HD treatments on patients, a balanced number of acupuncture sessions for the participants was determined.

#### 2.3.1. Verum Acupuncture Group

The experimental group (verum acupuncture) was divided into verum acupuncture subgroups A (3 × 3) and B (1 × 9). A total of 9 acupuncture treatments were performed, and the same choice of acupuncture points were applied to each subgroup. 

For each session and subject, a total of 8 needle insertions were made at five acupoints, namely Taixi (KI3), Sanyinjiao (SP6), Zusanli (ST36), Shenmen (HT7), and Guanyuan (CV4). Taixi (KI3), Sanyinjiao (SP6), and Zusanli (ST36) were punctured bilaterally, and Guanyuan (CV4) unilaterally. Additionally, Shenmen (HT7) was used unilaterally in the arm without the arterial-venous fistula (AVF) or in the right arm in those participants with a central venous catheter (CVC).

Sterilized and disposable stainless-steel needles (0.25 × 25 mm) were inserted and, after De qi sensation, were manually manipulated for one minute every ten minutes during needle retention (25 min). Interaction between the patient and the TCM practitioner has been kept to the minimum necessary to avoid non-specific treatment effects.

#### 2.3.2. Sham Acupuncture Group

The placebo group, which received sham acupuncture, was also divided into subgroups and received a total of nine acupuncture treatments. Manual acupuncture was performed using the same type of needles as described above, but with superficial needling (5 mm depth) at non-acupuncture points without attempting to achieve De qi sensation or stimulation. 

The locations of the acupuncture and non-acupuncture points referred to above are detailed in a previous published paper [32]. 

#### 2.3.3. Waiting List Group

Regarding the waiting-list group, no acupuncture treatment was performed from the time of randomization until the end of the follow-up period.

While each weekly HD session was underway, a licensed TCM specialist with five years of professional experience administered acupuncture treatment. The study timeline followed the standard care routine for HD sessions, and no other interventions were allowed.

### 2.4. Outcome Measurement

The primary outcome was the specific effect of acupuncture compared to sham acupuncture. This was observed through changes in HRQOL assessed using the physical component summary score (PCS) from the validated Portuguese version of the Kidney Disease Quality of Life—Short Form, Version 1.3 (KDQOL-SF^TM^ 1.3) [39]. Secondary outcomes included select multi-item scale and burden of kidney disease scale scores from the KDQOL-SF^TM^ 1.3.

The assessments were performed at baseline, after treatment, and at 12-week follow-up. 

#### 2.4.1. Kidney Disease Quality of Life—Short Form, Version 1.3 (KDQOL-SF^TM^ 1.3)

KDQOL-SF^TM^ 1.3 is a self-reported, disease-specific HRQOL measure composed of a generic scale including a 36-item health survey and a specific scale targeting the particular concerns of individuals with kidney failure with KRT (peritoneal dialysis or hemodialysis) [40]. 

The 36-Item Short Form Survey (SF-36) evaluates general health across eight different domains: physical functioning (10 items), role-physical (4 items), pain (2 items), general health (5 items), emotional well-being (5 items), role-emotional (3 items), social function (2 items), and energy/fatigue (4 items). Each question is scored from 0 (representing the poorest health) to 100 (indicating the best health). Physical and mental functioning are assessed by calculating normalized scores from the individual scales, referred to as physical component summary (PCS) and mental component summary (MCS) scores [41].

As a kidney disease-specific instrument, the KDQOL-SFT^TM^ 1.3 comprises 43 kidney items targeting kidney disease and assesses 11 kidney disease-specific components of HRQOL. These components include a symptom/problem list (12 items), effects of kidney disease (8 items), work status (2 items), burden of kidney disease (4 items), cognitive function (3 items), quality of social interaction (3 items), sexual function (2 items), sleep (4 items), social support (2 items), dialysis staff encouragement (2 items), and patient satisfaction (1 item). It also includes a single-item overall rating of health [40,41].

#### 2.4.2. Sociodemographic and Clinical Data Collection Form

This document was designed to collect patients’ sociodemographic information, such as gender, age, level of education, and employment status, as well as medical variables, including time on hemodialysis and laboratory data at baseline. The participant timeline is outlined in Table 1.

### 2.5. Statistical Analysis and Sample Size

For baseline, the demographic, clinical, and laboratorial data are presented by group through their means, standard deviations, and frequencies. To compare groups, Fisher’s exact test was used for categorical variables and a one-way ANOVA was used for continuous variables.

Due to the non-normality of the data (QL scores), non-parametric tests were used to analyze between-group differences. The Mann–Whitney and Kruskal–Wallis tests were used, followed by multiple comparison tests with Bonferroni correction. Within-group changes (baseline versus after treatment and baseline versus follow-up) were analyzed with the Wilcoxon signed-rank test. Sample size calculations for the RCT have been described previously [32].

The IBM SPSS Statistics software—version 27.0 [42] was used for statistical analysis. The statistical tests were performed at a significance level of 5%. 

## 3. Results

Between December 2020 and May 2021, 88 patients receiving maintenance HD were screened. After applying the inclusion criteria, 10 participants were excluded, 6 declined to participate, and a total of 72 patients were included and randomly assigned to the study groups. During the interval between the post-treatment assessment and the 12-week follow-up evaluation, one participant in the VA group, two in the SA group, and two in the WL group dropped out as a result of hospitalization or transplantation. As a result, the complete analysis set consisted of 67 patients, as shown in the trial flow diagram (Figure 1) [32]. Although the sample and methodology were the same as those used in the primary research project, the results described in the present study are from a new and different analysis conducted to evaluate the selected dimensions of HRQOL through KDQOL-SF^TM^ 1.3.

The baseline sociodemographic, clinical, and laboratorial characteristics of the overall population categorized by group are presented in Table 2. As previously reported [32], there were no statistically significant differences between the groups in sociodemographic, clinical (including the vascular access type), or laboratory variables (*p* > 0.05).

As the study sample mainly included older (mean age = 71.6; SD = 7.7) and retired people (85.1%), the KDQOL-SF^TM^ 1.3 subscales “Work status” and “Sexual function” were not considered for analysis. Additionally, “Social support,” “Dialysis staff encouragement,” and “Patient satisfaction” were not analyzed because the research team considered the subscales’ scores to not be dependent on the acupuncture protocol developed and applied.

The selected HRQOL dimension scores, when compared with the frequency of acupuncture treatment (3 treatments per week for 3 weeks vs. 1 treatment per week for 9 weeks), showed that changes at baseline vs. after treatment and at baseline vs. 12-week follow-up did not differ significantly in either the verum acupuncture group (VA) or the sham acupuncture group (SA) (*p* > 0.05), as shown in Table 3.

Since no significant differences were found among the treatment frequency groups, subgroups A and B were combined to compare VA, SA, and WL groups. Subsequently, a new statistical analysis was conducted, excluding the factor treatment frequency, as shown in Table 4 and Figure 2. These illustrate the effects of the group interventions at each time point in terms of the differences between baseline, after-treatment, and 12-week follow-up assessments of HRQOL dimensions.

No baseline differences between the groups (*p* > 0.05) were observed. As for the changes between baseline and after treatment, the scores of the KDQOL-SF^TM^ subscales symptom/problem list, effects of kidney disease, cognitive function, sleep, overall health, PCS, and MCS increased significantly in the VA group (*p* < 0.05) but not in the SA or WL groups (*p* > 0.05). Significant differences between groups (*p* < 0.05) were found in all these HRQOL dimensions, except in the MCS dimension (*p* = 0.415). The scores of burden of kidney disease and quality of social interaction did not change significantly in any of the groups (*p* > 0.05). 

Regarding the changes from baseline to follow-up, there were no statistically significant differences between groups (*p* > 0.05). However, scores of the sleep subscale (*p* = 0.026) decreased significantly in the VA group (*p* < 0.01) but not in the SA or WL groups (*p* > 0.05); the symptom/problem list and effects of kidney disease scores did not change in the VA and WL groups (*p* > 0.05), but decreased in the SA group (*p* < 0.05); the PCS score did not change in the VA group (*p* > 0.05), but decreased in the SA and WL groups (*p* > 0.05); the overall health score did not change in the VA and SA groups (*p* > 0.05), but decreased in the WL group (*p* < 0.05). The burden of kidney disease, cognitive function, and MCS scores did not change between baseline and follow-up in any of the groups (*p* > 0.05). The quality of social interaction score decreased significantly in all three groups (*p* < 0.05).

To assess the success of the blinding, patients were asked which type of acupuncture treatment they believed they had received (VA or SA). Most patients answered that they did not know (70% in the VA group and 68% in the SA group), and 31% believed they had received verum acupuncture (30% in the VA group and 32% in the SA group)—none of the patients believed he or she had received sham acupuncture treatment. The blinding index (0.84–95% CI: 0.78–0.91) shows that participant blinding was successful [32].

The absence of any reported unfavorable incidents by patients, caregivers, or physicians suggests that applying acupuncture throughout the HD sessions was harmless.

## 4. Discussion

Advanced CKD leads to kidney failure and often requires KRT. HD is a demanding treatment that is time-consuming for patients and limits their professional, family, and social activities. In addition, HD can cause both physical and emotional distress, which negatively impacts quality of life [6,43,44]. 

In this study, new data not described in the primary analysis of our previously conducted RCT [32] were analyzed to verify the effectiveness of acupuncture on HRQOL improvement. For the assessment of the HRQOL dimensions, the KDQOL-SF^TM^ 1.3 was chosen since it is a validated instrument for use on CKD patients [40,41] that includes a generic core that has been widely used as a measure of quality of life. 

In comparison with sham acupuncture or non-intervention, verum acupuncture improved the HRQOL after-treatment, as shown by the increase in KDQOL-SF^TM^ 1.3, physical component summary (PCS) scores, which included physical functioning, role functioning/physical, bodily pain, general health, vitality, and social functioning. The after-treatment results are consistent with those obtained in our previous study [32], where acupuncture improved functional capacity and peripheral muscle strength in patients undergoing HD. 

As regards the frequency of treatment, and in line with a prior study [32] in which no significant differences were observed, indicating that three treatments per week does not seem to provide better results than one treatment per week on functional capacity and muscle strength in patients undergoing HD, the outcomes of the present study also did not depend on the acupuncture treatment frequency. Therefore, three acupuncture treatments per week did not seem to result in a higher HRQOL compared to one treatment per week.

Some recent studies have explored the influence of acupuncture frequency on other clinical conditions, and the results are inconclusive and lack consistency. For instance, a study reported that three acupuncture sessions per week was more effective in treating knee osteoarthritis compared to one session per week, with improvements persisting for at least sixteen weeks [45]. In addition, a pilot study for postprandial distress syndrome showed that three sessions per week tended to improve symptoms and quality of life more than once a week after four weeks of treatment [46]. Finally, another study has shown that acupuncture treatment for lumbar disc herniation was equally effective, whether administered once every day or every two days. Both were more effective than receiving treatment once every three days [47]. Regardless, based on the results of the present study, it cannot be concluded that a higher frequency of acupuncture treatments leads to more significant therapeutic effects, and further research with robust results is needed to confirm the findings of this study.

Besides increasing PCS scores, verum acupuncture also increased the scores of some kidney disease-targeted areas (symptom/problem list, effects of kidney disease, cognitive function, and sleep), the overall health score, and the mental component summary (MCS) score, which is indicative of improvement in HRQOL. Sleep and cognitive function are two key components of HRQOL in kidney patients. Improved sleep quality may be reflected in a better mood and social interaction. On the other hand, cognitive decline is frequent in advanced kidney disease, removing the patient from reality and reducing his autonomy. Thus, the improvement of these two clinical parameters with acupuncture may contribute to the improvement of a patient’s global condition. 

Verum acupuncture did not improve the scores of burden of kidney disease or quality of social interaction. Although short-term improvement was observed, the long-term effects were not sustained in the VA group, as none of the HRQOL dimensions showed significant improvement three months after the intervention.

As for the specific effect of acupuncture compared to placebo (sham acupuncture), the results obtained are supported by earlier RCT’s in other medical conditions that have demonstrated the superiority of manual acupuncture over sham acupuncture and usual care in preventing episodic migraine without aura [48]. In addition, true acupuncture has been shown to be more effective in reducing joint pain in postmenopausal women with early-stage breast cancer and aromatase inhibitor-related arthralgias compared to sham acupuncture or waitlist control [49]. Moreover, acupuncture has been found to be more effective than sham acupuncture in increasing the response rate and elimination rate of all three cardinal symptoms (postprandial fullness, upper abdominal bloating, and early satiation) in patients with postprandial distress syndrome [46].

As far as the authors know, there is currently no RCT assessing the effectiveness of acupuncture in improving HRQOL in patients receiving maintenance HD. Bullen et al. (2018) conducted a study to assess the impact of acupuncture and massage on health-related quality of life during HD, observing a tendency towards overall HRQOL improvement [30]; Rehman et al. (2021) conducted a study to investigate the effectiveness of zolpidem 10 mg and acupressure therapy on foot acupoints and verified an improvement in both sleep quality and overall quality of life among HD patients experiencing CKD-associated pruritus [50]; Yıldırım Keskin and Taşci (2021) showed that acupressure had a beneficial impact on patients receiving HD treatment by increasing the amount of saliva generated, reducing the severity of visual analog scale thirst and positively affecting the HRQOL physical component sub-dimension [51]. Owing to methodological limitations, the use of TCM practices other than acupuncture and the heterogeneity of the instruments used to assess the quality of life, the results of this study cannot be compared.

Evidence from recent research showed the effect of acupuncture or electroacupuncture on HRQOL for those suffering different medical conditions. A systematic review and meta-analysis conducted by Hsieh et al. (2019) suggested that acupuncture therapy improves HRQOL in patients under medical treatment with chronic obstructive pulmonary disease [26]. Bao et al. (2021) assessed the impact of acupuncture on HRQOL outcomes in patients with solid tumors with chemotherapy-induced peripheral neuropathy, where statistically significant improvements in quality of life, anxiety, insomnia, and fatigue were achieved [52]. A pilot study conducted by Zhu et al. (2022) showed promising effects of electro-acupuncture in improving HRQOL, controlling symptom burden, and reducing toxicity during adjuvant chemotherapy in gastric cancer patients [27].

Low HRQOL scores in patients receiving maintenance HD have been associated with a higher risk of hospitalization and mortality [10,53,54,55], poorer mental health [56] and nutritional status [57], fatigue [58], chronic pain [59], and depressive and anxiety symptoms [44,60,61].

The findings of this study are promising as acupuncture proved to be a safe practice, contributing positively to PCS scores and improving the HRQOL in the studied clinical population. As the PCS score is associated with a more impaired functional status [62], an increase in this HRQOL parameter could mean an important contribution to the reduction of adverse outcomes or mortality related to the progression of CKD. The specific effect of acupuncture compared to sham acupuncture is reinforced by the results obtained in the previous RCT [32], in which increased functional capacity and peripheral muscle strength were also reported. Therefore, the results may also support clinicians considering the adoption of integrative and non-pharmacological practices for appropriate interventions in order to improve the HRQOL in patients undergoing HD.

Some limitations can be assigned to this study. In addition to those already widely discussed in the primary research [32], namely the sample size, the non-blinding of the TCM practitioner who provided the acupuncture treatment, and the insufficient follow-up time to assess the long-term effects of acupuncture, this study relied only on the analysis of a patient-reported outcome measure for its primary outcome. Additionally, the assessment of other HRQOL-related variables of the clinical population under study could have been considered, such as sleep quality, anxiety, and depression symptoms.

Based on the results presented here, future research is required to evaluate the efficacy of the acupuncture protocol employed in the study and confirm its clinical benefits in a larger sample of CKD patients receiving maintenance HD. It would also be interesting to compare this group with CKD patients on peritoneal dialysis over a longer follow-up period. Further studies should address the effects of acupuncture on HRQOL, also considering the complete assessment of mental health status and sleep quality. In order to validate integrative approaches in the care provided to CKD G5 patients, researchers might consider incorporating an active control group into the design of a new study. For instance, the effects of acupuncture could be compared with other TCM practices, such as acupressure or auricular acupuncture.

## 5. Conclusions

The use of integrative medicine practices may be a promising approach to improve HRQOL in CKD G5 patients undergoing HD. 

Compared to the sham acupuncture or no-acupuncture groups, the verum acupuncture group exhibited increased PCS scores, MCS scores, overall health, and specific kidney disease-targeted areas (symptom/problem list, effects of kidney disease, cognitive function, sleep). These positive effects were observed in the short term but did not persist three months after treatment.

Although acupuncture treatment tends to improve overall HRQOL, further investigation will be needed to validate the results of this study.

## Figures and Tables

**Figure 1 healthcare-11-01355-f001:**
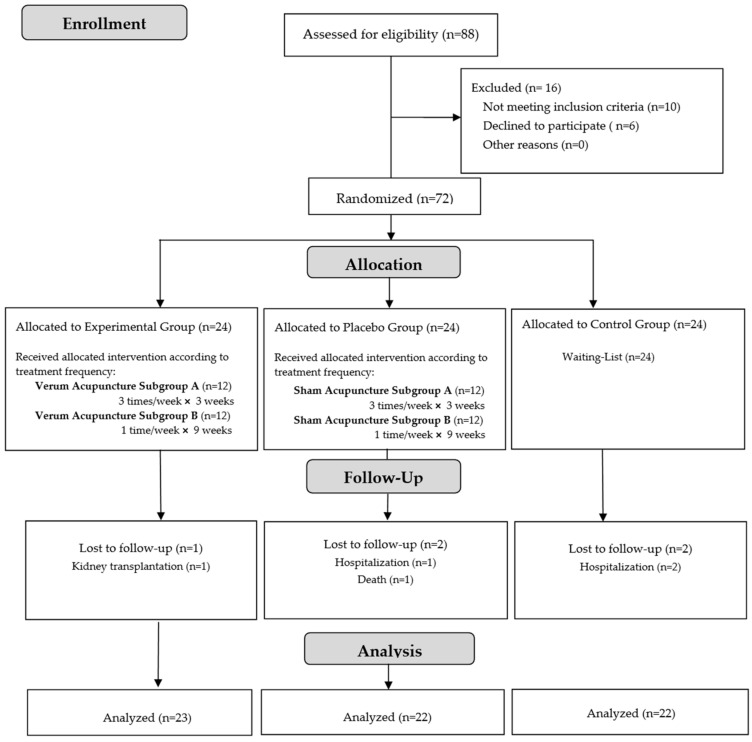
Trial flow diagram. Reprinted with permission from Ref. [32].

**Figure 2 healthcare-11-01355-f002:**
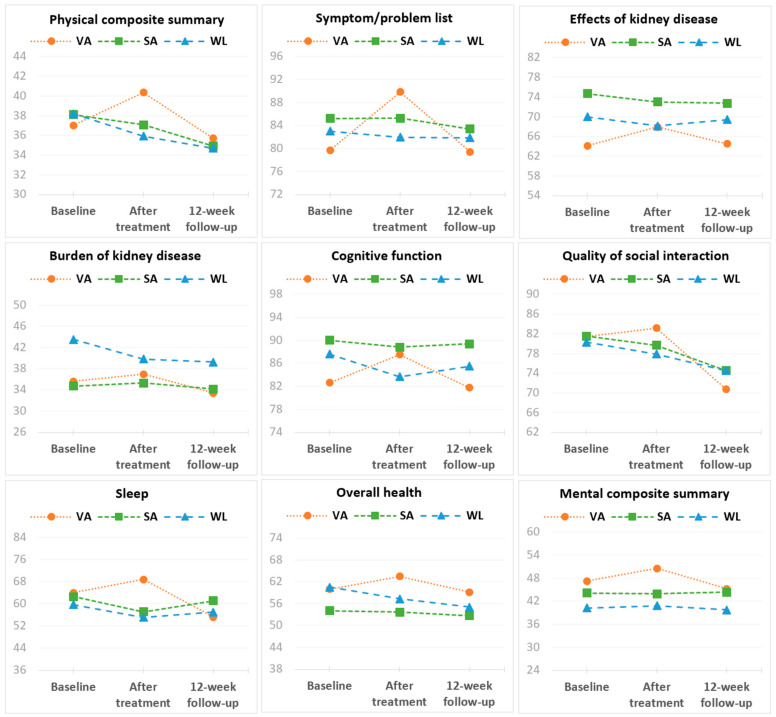
Mean HRQOL dimension scores at baseline, after treatment, and 12-week follow-up for Verum Acupuncture (VA), Sham Acupuncture (SA), and Waiting-List (WL) groups.

**Table 1 healthcare-11-01355-t001:** Timing of visits and data collection.

		Screening	Baseline	Treatment			Follow-Up after Treatment
			0-Week	1-Week	3-Week	9-Week	12-Week
**Patient**	Eligibility	X					
	Informed Consent		X				
	Demographics and clinical data		X				
	Physical examination		X				
	Randomization		X				
	Laboratorial data		X				X
**Intervention**	Verum Acupuncture Subgoup A (VA SgA)			9 Acupuncture treatments (3 sessions per week for 3 weeks)			X
	Verum Acupuncture Subgroup B (VA SgB)			9 Acupuncture treatments (1 session per week for 9 weeks)			
**Comparison**	Sham Acupuncture Subgroup A (SA SgA)			9 Acupuncture treatments in non-acupuncture points (3 sessions per week for 3 weeks)			X
	Sham Acupuncture Subgroup B (SA SgB)			9 Acupuncture treatments in non-acupuncture points (1 session per week for 9 weeks)			X
	Waiting-List Group (WL)			Non-Acupuncture Treatment			X
**Outcomes**	Health-related Quality of Life (Kidney Disease Quality of Life-Short Form, Version 1.3–KDQOL–SF^TM^ 1.3)		X		X	X	X
	Assessment of blinding success				X	X	
**Participant Safety**	Adverse effects			X	X	X	X

**Table 2 healthcare-11-01355-t002:** Sample baseline sociodemographic, clinical, and laboratorial characteristics overall by group.

Variables	Total (*n* = 67)	Verum Acupuncture (VA) Group (*n* = 23)	Sham Acupuncture (SA) Group (*n* = 22)	Waiting-List (WL) Group (*n* = 22)	*p*
Sociodemographic					
Gender					
Female	26 (38.8%)	9 (39.1%)	8 (36.4%)	9 (40.9%)	*1.000 ^(1)^*
Male	41 (61.2%)	14 (60.9%)	14 (63.6%)	13 (59.1%)	
Age					
Minimum–Maximum	56–91	60–84	57–91	56–87	
Mean (SD)	71.6 (7.7)	71.2 (5.1)	72.6 (8.3)	71.0 (9.4)	*0.764 ^(2)^*
Level of education					
No literacy	5 (7.5%)	0 (0.0%)	2 (9.1%)	3 (13.6%)	*0.279 ^(1)^*
1° Cycle (4 years)	49 (73.1%)	20 (87.0%)	17 (77.3%)	12 (54.5%)	
2° Cycle (6 years)	7 (10.4%)	2 (8.7%)	1 (4.5%)	4 (18.2%)	
3° Cycle (9 years)	1 (1.5%)	0 (0.0%)	0 (0.0%)	1 (4.5%)	
High school (12 years)	5 (7.5%)	1 (4.3%)	2 (9.1%)	2 (9.1%)	
Employment status					
Employed	4 (6.0%)	0 (0.0%)	2 (9.1%)	2 (9.1%)	*0.481 ^(1)^*
Self-employed	4 (6.0%)	2 (8.7%)	0 (0.0%)	2 (9.1%)	
Unemployed	2 (3.0%)	1 (4.3%)	1 (4.5%)	0 (0.0%)	
Retired	57 (85.1%)	20 (87.0%)	19 (86.4%)	18 (81.8%)	
Clinical					
Hemodialysis time					
<12 months	2 (3.0%)	2 (8.7%)	0 (0.0%)	0 (0.0%)	*0.240 ^(1)^*
12 to 120 months	56 (83.6%)	19 (82.6%)	20 (90.9%)	17 (77.3%)	
>120 months	9 (13.4%)	2 (8.7%)	2 (9.1%)	5 (22.7%)	
Vascular access					
Arteriovenous fistula (AVF)	61 (91.0%)	22 (95.7%)	18 (81.8%)	21 (95.5%)	*0.306 ^(1)^*
Central venous catheter (CVC)	6 (9.0%)	1 (4.3%)	4 (18.2%)	1 (4.5%)	
Laboratorial	Mean (SD)				
Hemoglobin (g/dL)	10.93 (1.03)	10.88 (0.94)	10.86 (1.05)	11.05 (1.12)	*0.797 ^(2)^*
Potassium (mEq/L)	5.44 (0.81)	5.52 (0.89)	5.35 (0.81)	5.45 (0.74)	*0.799 ^(2)^*
Calcium (mg/dL)	9.11 (0.51)	9.13 (0.55)	9.10 (0.49)	9.10 (0.51)	*0.970 ^(2)^*
Phosphorus (mg/dL)	4.74 (1.13)	4.71 (1.17)	4.70 (1.25)	4.80 (1.00)	*0.944 ^(2)^*
Sodium (mg/dL)	138.1 (3.0)	138.5 (2.9)	138.6 (3.8)	137.3 (2.0)	*0.296 ^(2)^*
Albumin (g/dL)	3.91 (0.28)	3.98 (0.27)	3.81 (0.27)	3.95 (0.29)	*0.101 ^(2)^*
Urea (before HD) (mg/dL)	152.6 (37.5)	161.0 (37.9)	145.3 (36.2)	151.0 (38.3)	*0.369 ^(2)^*
Creatinine (mg/dL)	9.94 (2.12)	10.15 (2.26)	9.51 (1.94)	10.16 (2.16)	*0.514 ^(2)^*
Parathyroid hormone (pg/mL)	418.0 (243.2)	454.2 (256.9)	358.2 (193.3)	439.9 (271.5)	*0.370 ^(2)^*
Cholesterol (mg/dL)	162.8 (37.1)	166.0 (47.2)	155.5 (35.8)	166.6 (25.0)	*0.536 ^(2)^*

^(1)^ significance value of Fisher’s exact test; ^(2)^ significance value of ANOVA. Reprinted with permission from Ref. [32].

**Table 3 healthcare-11-01355-t003:** Comparison of the changes in HRQOL dimension scores between treatment frequency, for Verum Acupuncture (VA) and Sham Acupuncture (SA) groups.

	Verum Acupuncture (VA) Group			Sham Acupuncture (SA) Group		
HRQOL Domains	Treatment Frequency 3 × 3 (*n* = 12)	Treatment Frequency 1 × 9 (*n* = 11)	*p*	Treatment Frequency 3 × 3 (*n* = 12)	Treatment Frequency 1 × 9 (*n* = 10)	*p*
Kidney disease targeted areas Symptom/problem list	**78.0 ± 15.6**	**81.6 ± 10.1**		**81.6 ± 10.6**	**89.6 ± 8.4**	
Baseline–after treatment	10.9 ± 8.4	9.3 ± 4.8	0.709	0.6 ± 3.3	−0.7 ± 1.9	0.214
Baseline–12-week follow-up	0.5 ± 4.5	−1.1 ± 3.3	0.281	−2.5 ± 2.5	−1.1 ± 2.9	0.768
Effects of kidney disease	**65.6 ± 15.9**	**62.5 ± 20.1**		**71.0 ± 24.0**	**79.8 ± 11.9**	
Baseline–after treatment	4.4 ± 4.3	3.1 ± 3.4	0.352	−0.5 ± 5.0	−3.1 ± 4.4	0.146
Baseline–12-week follow-up	1.0 ± 4.3	−0.3 ± 5.5	0.296	−2.3 ± 2.7	−1.6 ± 2.2	0.833
Burden of kidney disease	**34.4 ± 17.4**	**36.9 ± 16.2**		**38.0 ± 31.8**	**30.6 ± 21.3**	
Baseline–after treatment	2.1 ± 6.2	0.6 ± 3.4	0.445	−0.6 ± 6.0	2.1 ± 7.9	0.335
Baseline–12-week follow-up	−1.0 ± 3.6	−3.4 ± 7.6	0.594	−1.6 ± 1.8	−0.3 ± 3.5	0.883
Cognitive function	**83.9 ± 19.2**	**81.2 ± 14.2**		**91.1 ± 11.1**	**88.7 ± 7.7**	
Baseline–after treatment	3.3 ± 6.7	6.7 ± 8.4	0.315	−1.7 ± 5.8	−0.7 ± 3.8	0.766
Baseline–12-week follow-up	−1.4 ± 2.8	−0.4± 3.1	0.158	−1.1 ± 2.6	0.0 ± 0.0	0.298
Quality of social interaction	**83.9 ± 18.1**	**78.8 ± 20.6**		**79.4 ± 17.2**	**84.0 ± 6.4**	
Baseline–after treatment	0.6 ± 9.6	3.2 ± 9.6	0.746	−2.8 ± 4.5	−0.7 ± 3.8	0.179
Baseline–12-week follow-up	−9.7 ± 8.7	−11.8 ± 19.0	0.924	−7.9 ± 14.9	−5.7 ± 3.2	0.800
Sleep	**65.0 ± 20.5**	**63.0 ± 21.2**		**57.9 ± 18.1**	**68.3 ± 12.8**	
Baseline–after treatment	4.0 ± 15.0	6.1 ± 8.4	0.615	−5.2 ± 10.1	−5.8 ± 9.7	0.840
Baseline–12-week follow-up	−7.5 ± 5.8	−10.5 ± 11.7	0.283	0.6 ± 5.4	−4.0 ± 18.7	0.642
Overall Health	**63.3 ± 17.8**	**56.4 ± 10.3**		**58.3 ± 18.5**	**49.0 ± 5.7**	
Baseline–after treatment	2.5 ± 7.5	4.5 ± 9.3	0.598	−1.7 ± 11.1	1.0 ± 5.7	0.393
Baseline–12-week follow-up	−1.7 ± 7.2	0.0 ± 6.3	0.677	−4.2 ± 13.8	2.0 ± 6.3	0.245
36-item health survey (SF-36) SF12-Physical composite	**37.8 ± 10.9**	**36.1 ± 7.8**		**37.3 ± 10.4**	**38.9 ± 12.4**	
Baseline–after treatment	3.1 ± 5.3	3.6 ± 4.4	0.518	−0.7 ± 3.1	−1.4 ± 3.2	0.509
Baseline–12-week follow-up	−0.9 ± 5.1	−1.7 ± 4.5	0.782	−2.3 ± 4.5	−4.2 ± 5.1	0.176
SF12-Mental composite	**50.8 ± 10.6**	**43.4 ± 10.3**		**44.3 ± 8.7**	**44.0 ± 7.3**	
Baseline–after treatment	3.9 ± 7.5	2.8 ± 6.9	0.601	−0.5 ± 3.0	0.2 ± 1.7	0.741
Baseline–12-week follow-up	−2.4 ± 3.8	−1.5 ± 6.5	0.406	−0.7 ± 3.2	1.4 ± 3.9	0.373

NOTES: (1) for each HRQOL dimension: the first row (bold) shows the mean (M) and standard deviation (SD) of the domain score, the second row shows the difference from baseline to after treatment, and the third row shows the difference from baseline to 12-week follow-up; (2) treatment frequency 3 × 3—3 treatments a week for 3 weeks, treatment frequency 1 × 9—1 treatment a week for 9 weeks; (3) *p*-value of Mann–Whitney test for comparison of the changes (baseline vs. after treatment and baseline vs. follow-up) between treatment frequency for the Verum Acupuncture group and the Sham Acupuncture group.

**Table 4 healthcare-11-01355-t004:** Description of HRQOL dimension scores and comparison of the changes from baseline to after treatment and from baseline to 12-week follow-up between groups.

HRQOL Domains/ KDQOL-SF^TM^ 1.3 Scores	Verum Acupuncture (VA) Group (*n* = 23)	Sham Acupuncture (SA) Group (*n* = 22)	Waiting-List (WL) Group (*n* = 22)	*p*
Primary Outcome				
36-item health survey (SF-36) scale				
Physical composite summary (PCS)	**37.0 ± 9.4**	**38.1 ± 11.1**	**38.2 ± 8.5**	**0.953**
Baseline–after treatment	3.34 ± 4.77 **	−1.02 ± 3.11 ^A^	−2.27 ± 5.61 *^A^	<0.001
Baseline–follow-up	−1.29 ± 4.72	−3.18 ± 4.73 *	−3.50 ± 5.03 *	0.283
Secondary Outcomes				
Kidney disease targeted areas				
Symptom/problem list	**79.7 ± 13.1**	**85.2 ± 10.3**	**83.0 ± 8.8**	**0.320**
Baseline–after treatment	10.14 ± 6.83 **	0.09 ± 3.18 ^A^	−1.04 ± 2.47 ^A^	<0.001
Baseline–12–week follow–up	−0.27 ± 3.99	−1.80 ± 2.67 *	−1.14 ± 3.20	0.409
Effects of kidney disease	**64.1 ± 17.6**	**74.7 ± 20.5**	**70.0 ± 19.4**	**0.151**
Baseline–after treatment	3.80 ± 3.88 **	−1.70 ± 4.80 ^A^	−1.85 ± 2.67 *^A^	<0.001
Baseline–12-week follow-up	0.41 ± 4.83	−1.99 ± 2.47 *	−0.57 ± 1.57	0.244
Burden of kidney disease	**35.6 ± 16.5**	**34.7 ± 27.2**	**43.5 ± 23.9**	**0.337**
Baseline–after treatment	1.36 ± 4.97	0.57 ± 7.19	−3.69 ± 9.18	0.084
Baseline–12-week follow-up	−2.17 ± 5.84	−0.57 ± 2.67	−4.26 ± 12.56	0.586
Cognitive function	**82.6 ± 16.7**	**90.0 ± 9.6**	**87.6 ± 12.6**	**0.431**
Baseline–after treatment	4.93 ± 7.58 **	−1.21 ± 4.88 ^A^	−3.94 ± 8.40 *^A^	0.001
Baseline–12-week follow-up	−0.87 ± 3.05	−0.61 ± 1.96	−2.12 ± 8.82	0.944
Quality of social interaction	**81.4 ± 19.1**	**81.5 ± 13.3**	**80.3 ± 15.2**	**0.789**
Baseline–after treatment	1.74 ± 9.69	−1.82 ± 4.21	−2.42 ± 13.58	0.096
Baseline–12-week follow-up	−10.72 ± 14.32 **	−6.97 ± 11.17 **	−5.76 ± 13.22 *	0.199
Sleep	**64.0 ± 20.4**	**62.6 ± 16.4**	**59.7 ± 16.4**	**0.488**
Baseline–after treatment	4.89 ± 12.69 *	−5.45 ± 9.69 *^A^	−4.55 ± 13.84 *^A^	0.001
Baseline–12-week follow-up	−8.80 ± 9.62 **	−1.48 ± 13.06 ^A^	−2.61 ± 13.53 ^A^	0.026
Overall Health	**60.0 ± 14.8**	**54.1 ± 14.7**	**60.5 ± 17.0**	**0.270**
Baseline–after treatment	3.48 ± 8.32 *^A^	−0.45 ± 8.99 ^AB^	−3.18 ± 7.16 *^B^	0.048
Baseline–12-week follow-up	−0.87 ± 6.68	−1.36 ± 11.25	−5.45 ± 9.12 *	0.084
36-item health survey (SF-36) scale				
Mental composite summary (MCS)	**47.2 ± 10.9**	**44.1 ± 7.9**	**40.2 ± 6.5**	**0.078**
Baseline–after treatment	3.35 ± 5.06 *	−0.18 ± 2.49	0.61 ± 2.48	0.415
Baseline–12-week follow-up	−1.98 ± 5.18	0.27 ± 3.64	−0.46 ± 2.40	0.228

NOTES: (1) for each HRQOL dimension: the first row (bold) shows the mean (M) and standard deviation (SD) of the domain score at the baseline, the second row shows the difference from baseline to after treatment, and the third row shows the difference from baseline to 12-week follow-up; (2) *p*-value of Kruskal–Wallis test for comparison between groups; ^A,B^ groups with a superscript letter in common do not differ significantly, *p* > 0.05 in the multiple comparison tests with Bonferroni correction; (3) within-group changes at baseline vs. after treatment and baseline vs. 12-week follow-up were assessed using Wilcoxon signed-rank test; * *p* < 0.05 and ** *p* < 0.01.

## Data Availability

Not applicable.

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
