# Peer review of "Effectiveness of Acupuncture on Health-Related Quality of Life in Patients Receiving Maintenance Hemodialysis"

_healthcare, 2023, doi:10.3390/healthcare11091355_

Round 1

Reviewer 1 Report

Generally, because this is an independent study the main parameters should not only be referenced from the prior study or the prior study protocol but should be included in the paper, so the reader finds these information easily.

The timeline (in weeks) should be clearly described in all figures and tables. 

Introduction: The authors should give more references and cite prior studies, which explain why acupuncture should be applied to patients undergoing hemodialysis. 

Methods: The authors should describe their acupuncture concept in detail and explain why they have chosen these two subgroups. In the verum acupuncture, 5 obligatory points and 3 optional points were used. Describes details regarding the optional points. Is it correct that the verum group got 8 points and the control group only 5 points?

Results: Subgroup results should be included in the paper, not only as supplementary data. 

Discussion: The authors should more critically discuss their data, especially why their subgroups did not show any differences. Sentences like "The impact of the frequency of acupuncture treatment has already been discussed in a previous paper". " are not adequate and should be newly discussed. 

Reviewer 2 Report

The paper is good. However, it needs some minor revisions:

- please describe more about the time of the acupuncture. When do you provide acupuncture? During the dialysis session or between the dialysis session. On the same day of dialysis or on other days? During the dialysis unit or other units?

- Did you check and compare groups based on vascular access methods? The vascular access of AVF may interfere with acupuncture and may impact the results. 

- Please provide more details about the blinding in the methods section.

- The conclusion needs revisions. The results show the short-term effects of Verum acupuncture. 

Reviewer 3 Report

Why were those who were unable to engage in physical exercise excluded?

Patients were only excluded if they had acupuncture less than 2 weeks prior. That is a very short wash-out period. Curious how many were recent acupuncture patients (maybe having had acupuncture in the preceding 3-6 months).

Per STRICTA - a rationale for why the acupuncture prescriptions was chosen would be nice. Why not a different point prescription?

A deeper discussion about the lack of a difference in outcomes between the 3 visits per week for three weeks or 1 visit per week for nine weeks would be of value. Frequency and duration (F&D) is an important question that isn't always discussed in the literature especially since most trial designs only have one treatment F&D. I think this should be mentioned in the abstract and conclusion because it is very useful, glad you chose to look at two different F&D approaches. 

Nicely done. 
